# Expanding the MAPPs Assay to Accommodate MHC-II Pan Receptors for Improved Predictability of Potential T Cell Epitopes

**DOI:** 10.3390/biology12091265

**Published:** 2023-09-21

**Authors:** Katharina Hartman, Guido Steiner, Michel Siegel, Cary M. Looney, Timothy P. Hickling, Katharine Bray-French, Sebastian Springer, Céline Marban-Doran, Axel Ducret

**Affiliations:** 1Roche Pharma Research and Early Development, Roche Innovation Center Basel, Grenzacherstrasse 124, 4070 Basel, Switzerlandmike.looney@roche.com (C.M.L.);; 2School of Science, Department of Biochemistry and Cell Biology, Constructor University, Campus Ring 1, 28759 Bremen, Germany

**Keywords:** anti-drug antibody, immunogenicity, *in silico* analysis, MAPPs assay, mass spectrometry, HLA II receptors, NetMHCIIpan, immunopeptidomics, T cell epitope, therapeutic antibodies

## Abstract

**Simple Summary:**

The major histocompatibility complex class II-associated peptide proteomics assay is widely used during preclinical immunogenicity risk assessments to identify biotherapeutic-derived peptides. These potential T cell epitopes are presented by dendritic cells and may trigger CD4^+^ T helper cell activation, which could lead to downstream anti-drug antibody secretion by plasma cells. Currently, the utility of this immunopeptidomics assay has often been restricted to studying human leukocyte antigen-DR receptors due to the lack of well-characterized human leukocyte antigen-DP, -DQ, and pan antibodies available. Here, we seek to accommodate major histocompatibility complex class II pan receptors by testing commonly commercially available antibody clones, and characterizing their specificity via the epitope prediction algorithm NetMHCIIpan. Although the application of these antibodies in the assay increased the identified compound-specific cluster profile, no individual antibody clone was able to recover the complete human leukocyte antigen II peptide repertoire. Our findings reveal that a mixed immunoprecipitation strategy utilizing a minimum of three antibody clones with differing specificities (human leukocyte antigen-DR-specific clone L243, pan-specific clone WR18, and -DQ-specific clone SPV-L3) leads to more robust compound-specific peptide detection in one single analysis. Ultimately, expanding the assay to leverage human leukocyte antigen pan receptors improves the predictability of additional potential T cell epitopes.

**Abstract:**

A critical step in the immunogenicity cascade is attributed to human leukocyte antigen (HLA) II presentation triggering T cell immune responses. The liquid chromatography–tandem mass spectrometry (LC-MS/MS)-based major histocompatibility complex (MHC) II-associated peptide proteomics (MAPPs) assay is implemented during preclinical risk assessments to identify biotherapeutic-derived T cell epitopes. Although studies indicate that HLA-DP and HLA-DQ alleles are linked to immunogenicity, most MAPPs studies are restricted to using HLA-DR as the dominant HLA II genotype due to the lack of well-characterized immunoprecipitating antibodies. Here, we address this issue by testing various commercially available clones of MHC-II pan (CR3/43, WR18, and Tü39), HLA-DP (B7/21), and HLA-DQ (SPV-L3 and 1a3) antibodies in the MAPPs assay, and characterizing identified peptides according to binding specificity. Our results reveal that HLA II receptor-precipitating reagents with similar reported specificities differ based on clonality and that MHC-II pan antibodies do not entirely exhibit pan-specific tendencies. Since no individual antibody clone is able to recover the complete HLA II peptide repertoire, we recommend a mixed strategy of clones L243, WR18, and SPV-L3 in a single immunoprecipitation step for more robust compound-specific peptide detection. Ultimately, our optimized MAPPs strategy improves the predictability and additional identification of T cell epitopes in immunogenicity risk assessments.

## 1. Introduction

A critical step in the immunogenicity cascade is the ability of dendritic cells (DCs) to present therapeutic monoclonal antibody (mAb)-derived peptides via major histocompatibility complex (MHC) class II receptors to naïve CD4^+^ T cells. Recognition and binding of the MHC-II receptor–peptide complexes to the T cell receptor (TCR) leads to the activation and proliferation of CD4^+^ T cells. In turn, T cell-dependent events are initiated, such as the downstream activation and proliferation of B cells. The B cells specific to the therapeutic mAb differentiate into either memory cells or anti-drug antibody (ADA)-secreting plasma cells [1,2,3,4].

The blocking of the antigen-binding site of a therapeutic mAb via ADA formation may reduce a drug’s effectiveness, render it completely ineffective, or even elicit toxicities in subjects treated [5,6,7,8,9]. Due to this, immunogenicity may have a major impact on the treatment of patients and their ability to obtain an efficacious and safe therapy [9,10,11,12,13,14]. Consequently, the imminent risks of generating an immune response after mAb administration need to be critically evaluated during drug development phases.

DCs process and present endogenous and exogenous antigens in the form of linear epitopes bound in the groove of MHC-II receptors [1,2,3]. In humans, HLA class II receptors are encoded by three different loci, HLA-DR, HLA-DQ, and HLA-DP [15,16,17]. Up to eight different HLA II heterodimers can be expressed in a heterozygotic individual (two DRB1/DRA, two DRB3/4/5/DRA, two DP, and two DQ) [15]. Polymorphism is a notable feature of MHC-II genes [18,19]. For example, the HLA-DRB locus accounts for more than 4300 known alleles at population level [20].

Studies have shown a difference in cell surface expression of the three main HLA II receptor families, in which there is a significantly higher (~5–10-fold) HLA-DR expression level in comparison to HLA-DP and HLA-DQ [15]. This correlates with the frequency distribution for HLA class II antigen-specific responses and number of epitopes in which ~80%, ~20%, and 5–10% of MHC-II-identified peptides are HLA-DR-, HLA-DP-, and HLA-DQ-specific, respectively [21,22]. Due to this, HLA-DR has been, and is still, the most intensively studied MHC-II isotype [19].

At population level, interpatient differences in HLA genotypes result in the expression in each individual of a small subset of a wide variety of HLA class II receptors differing in binding affinities. This results, not unexpectedly, in the observation of various rates of immunogenicity toward specific therapeutic mAbs [1,5]. In addition, recent data demonstrate that the “minor” HLA-DP and HLA-DQ alleles also contribute to triggering a significant CD4^+^ T cell response [15]. For example, the presence of the HLA-DQA1*05 allele, which is carried by approximately 40% of Europeans, has been shown to significantly increase the rate of immunogenicity in patients treated with adalimumab [23]. Overall, this indicates that an accurate preclinical drug immunogenicity assessment must integrate comprehensive *in silico-* and *in vitro*-based tools taking into account the HLA diversity encountered in the general population [21].

The experimental identification of MHC-II-presented therapeutic mAb-derived peptides, signifying T cell epitopes, is accomplished through the liquid chromatography-tandem mass spectrometry (LC-MS/MS)-based MHC-II-associated peptide proteomics (MAPPs) assay. Mechanistically, the MAPPs assay provides insights into what may occur in vivo through the identification of potential T cell epitopes presented by HLA class II receptors after uptake and processing of the full-length mAb by DCs [3,24,25].

Typically, the MAPPs assay is performed using monocyte-derived dendritic cells (moDCs) as specialized antigen-presenting cells (APCs). After maturation and challenge with the compound of interest, MHC-II peptide–receptor complexes are first immune-precipitated using a suitable reagent, typically an antibody specific for HLA class II receptors. Bound MHC-II peptides are then acid-eluted from the receptor complexes and characterized by LC-MS/MS [21].

Currently, most MAPPs studies have been restricted to the identification of HLA-DR-presented peptides, mostly due to well-performing (in terms of binding specificity) immunoprecipitating pan HLA-DR antibodies, such as the mAb clone L243. The lack of well-characterized immunoprecipitating HLA-DP and HLA-DQ antibodies [21] has so far prevented the generation of corresponding datasets for immunogenicity risk assessments. In recent years, however, the increased speed and sensitivity of modern mass spectrometers has enabled the identification of less abundant MHC-II peptides, such as those that could be presented by HLA-DP and HLA-DQ receptors. Thus, analytically, a comprehensive analysis of complex peptide mixtures generated using more general immunoprecipitating antibodies can now be considered.

To our knowledge, few studies have reported the use of HLA-DQ and/or MHC-II pan antibodies in a MAPPs assay to identify potential T cell epitopes [26,27,28]. Furthermore, although a number of HLA-DP, HLA-DQ, and MHC-II pan-specific receptor-precipitating antibodies exist, to our knowledge, the specificity and profile of such reagents—including the conventional HLA-DR antibody clone L243—in various immunoprecipitation (IP) strategies have not been characterized systematically according to peptide binding affinity.

Here, we present the comparison of multiple commercially available MHC-II antibody clones, such as our gold-standard HLA-DR antibody (L243) versus HLA-DQ (SPV-L3 and 1a3), HLA-DP (B7/21), and MHC-II pan (Tü39, CR3/43, and WR18) antibody clones in individual and various combinations of mixed IP strategies in the MAPPs assay. Identified MHC-II receptor-derived peptides are subsequently characterized by the HLA-II epitope-prediction algorithm NetMHCIIpan [29] to assess the binding capacity of the enriched MHC-II peptides to HLA-DR, HLA-DP, and HLA-DQ alleles and profile antibody immunoprecipitating characteristics. Moreover, since studies have asserted the implications of certain HLA-DR and HLA-DQ alleles with respect to immunogenicity in response to therapeutic mAb treatment [23,30,31], we aim to assess the combination(s) of immunoprecipitating antibodies enabling the identification of therapeutic mAb-specific HLA-DQ and HLA-DP receptor-derived peptides. We also compared the MHC-II antibody clones in single or mixed IP procedures to determine a format that is compatible with screening biotherapeutic candidates during preclinical immunogenicity risk assessments for the identification of biotherapeutic-derived MHC-II pan-specific potential T cell epitopes. The advantage of expanding the MAPPs assay to leverage HLA-DP and HLA-DQ receptors would vastly improve the predictability of immunogenicity through the identification of a greater number of potential T cell epitopes during preclinical drug development.

## 2. Materials and Methods

### 2.1. Antibodies and Compounds

The fully human anti-interleukin (IL)-21 receptor (ATR-107) mAb was kindly gifted from Genentech. The marketed fully human mAb adalimumab (Humira^®^, Abbvie, Chicago, IL, USA) was included as a clinical grade benchmark product purchased from a pharmacy. ATR-107 and adalimumab were stored at −80 °C and 4 °C, respectively. Stock solutions of the highly immunogenic antigen keyhole limpet hemocyanin (KLH-Imject Maleimide-Activated mcKLH; Thermo Fisher Scientific Inc, Waltham, MA, USA; Cat: #77600), which served as a positive control, were reconstituted in sterile water at 10 mg/mL and stored at 4 °C.

### 2.2. Human Donors

Cryopreserved peripheral blood mononuclear cells (PBMCs) from genotyped healthy human donors were provided by Lonza (Lonza Group AG; Basel, Switzerland), which were obtained from consenting anonymous donors in accordance with current ethical practices.

### 2.3. MHC-II-Associated Peptide Proteomics (MAPPs) Assay

#### 2.3.1. Generation of Monocyte-Derived Dendritic Cells (moDCs)

CD14^+^ mononuclear cells were purified from PBMCs via CD14 magnetic microbeads (Miltenyi Biotec, Bergisch, Gladbach, Germany; Cat: #130-050-201) using the QuadroMACS system (Miltenyi Biotec, Bergisch, Gladbach, Germany; Cat: #130-090-976) according to the manufacturer’s recommendations. Cells were cultured at a density of 0.3 × 10^6^ cells/mL in warm CellGenix DC medium (Sartorius CellGenix GmbH, Freiburg, Germany; Cat: #20801-0500) supplemented with 1% (*v/v*) Glutamax (Gibco Thermo Fisher Scientific Inc., Waltham, MA, USA; Cat: #35050-061), 1% (*v/v*) non-essential amino acids (Gibco; Cat: #11140-035, 1% (*v/v*) sodium pyruvate (Gibco Thermo Fisher Scientific Inc., Waltham, MA, USA; Cat: #11360-039), 1% (*v/v*) penicillin–streptomycin (Gibco Thermo Fisher Scientific Inc., Waltham, MA, USA; Cat: #15140-122). CD14^+^ cells were plated at 2.5 × 10^6^ cells per sample in 25 cm^2^ ultra-low attachment culture flasks (Corning Inc., Corning, NY, USA; Cat: #4616), and differentiated into immature moDCs using 5 ng/mL recombinant human IL-4 (R&D Systems, Minneapolis, MN, USA; Cat: #204-IL) and 50 ng/mL recombinant human GM-CSF (R&D Systems, Minneapolis, MN, USA; Cat: #215GM-500) for 5 days at 37 °C in an atmosphere of 5% CO_2_.

#### 2.3.2. Loading, Maturation, and Lysis of moDC

Immature moDCs were challenged with either adalimumab (0.3 μM), ATR-107 (0.3 μM), or the positive control KLH (50 μg/mL), and matured with lipopolysaccharide (LPS; 1 μg/mL) from *Salmonella enterica* (Sigma Aldrich, St. Louis, MO, USA; Cat: #L5886) for 24 h at 37 °C and 5% CO_2_. After harvesting, moDCs were lysed in a hypotonic buffer (20 mM Tris-HCl pH 7.8; 5 mM MgCl_2_) containing 1% (*v*/*v*) Triton X-100 Surfact-Amps detergent solution (Thermo Fisher Scientific Inc., Waltham, MA, USA; Cat: #28314) and a protease inhibitor mini tablet (Thermo Fisher Scientific Inc., Waltham, MA, USA; Cat: #A32955) for 1 h in a ThermoMixer (Eppendorf, Hamburg, Germany; Cat: #5355000.011) at 1100 rpm and 4 °C. Following centrifugation for 10 min at 14,000 rpm and 4 °C, lysates were collected and frozen at −80 °C.

#### 2.3.3. Immunoprecipitation of HLA II Receptors and Elution of MHC-II-Specific Peptides

Lysates were incubated with either individual or mixed preparations of clone L243 (10 μg; anti-human HLA-DR biotin; RayBiotech, Peachtree Corners, GA, USA; Cat: #150-10306), clone SPV-L3 (15 μg; anti-human HLA-DQ biotin; Biotium, Fremont, CA, USA; Cat: #BNCB0200-500), clone 1a3 (15 μg; anti-human HLA-DQ biotin; Leinco Technologies, Fenton, MO, USA; Cat: #H136-200ug) clone B7/21 (15 μg; anti-human HLA-DP biotin; Leinco Technologies, Fenton, MO, USA; Cat: #H254-200 µg), clone CR3/43 (15 μg; anti-human MHC-II pan biotin; Bio-Techne, Minneapolis, MN, USA; Cat: #NBP2-54507B), clone WR18 (15 μg; anti-human MHC-II pan biotin; Bio-Techne, Minneapolis, MN, USA; Cat: #NB100-64358B), or clone Tü39 (15 μg; anti-human MHC-II pan; biotin conjugation by Biolegend, San Diego, CA, USA; Cat: #361702) overnight at 4 °C on a rotator. Immunoprecipitation of MHC-II receptors was carried out using the automated AssayMAP Bravo platform (Agilent Technologies, Santa Clara, CA, USA, Cat: #3029078) and accompanying streptavidin cartridges (Agilent Technologies, Santa Clara, CA, USA, Cat: #G5496-60021). Following priming (1% (*v/v*) formic acid; 100 mM NaCl) and equilibration (20 mM Tris-HCl pH 7.8; 5 mM MgCl_2_), samples were loaded onto the cartridges and washed in buffer (20 mM HEPES pH 7.9; 150 mM KCl; 1 mM MgCl_2_; 0.2 mM CaCl_2_; 0.2 mM EDTA; 10% (*v/v*) glycerol; 0.1% (*v/v*) NP-40 alternative (MilliporeSigma, Burlington, MA, USA; Cat: #492018-50 mL)) and Milli-Q H_2_O. HLA II-specific peptides were eluted in 0.1% (*v/v*) TFA in a final volume of 18 μL. Peptide samples were then directly loaded onto Evosep C18 tips (Evosep Biosystems, Odense, Denmark; Cat: #EV2001) according to the manufacturer’s recommendations and stored at 4 °C until LC-MS/MS analysis.

#### 2.3.4. LC-MS/MS Method

Peptide samples were analyzed in a trapped ion mobility time-of-flight mass spectrometer (TimsTOF PRO 2, Bruker Daltonics, Bremen, Germany) equipped with a captive electrospray source operated at 1200–1400 V. Peptides were separated via reverse-phase chromatography using an Aurora Elite column (15 cm × 75 μm i.d., 1.7 μm particle size, heated at 45 °C; Ion Opticks) using an Evosep One standardized nanoLC platform (Evosep, Odense, Denmark; EV-1000). A turnaround time of 31 min was achieved using the Evosep’s built-in program 40 SPD Whisper. Eluted MHC-II peptides were analyzed using data-directed analysis following standard operating parameters. The TIMS accumulation/ramping time was set to 150 ms (mobility range: 0.6–1.6) while the TOF analyzer was set to record ions in the mass range m/z 100–1700 (global cycle time: 1.71 s). One survey scan (selection range for MS/MS analysis: *m/z* 350–1300, ion mobility 0.7–1.5, 1 < z < 6) was followed by MS/MS analysis in PASEF mode including up to 10 TIMS ramps per full cycle. Dynamic exclusion prevented the repeated selection of an ion for MS/MS analysis for 9 s.

#### 2.3.5. LC-MS/MS Data Analysis

The LC-MS/MS raw data files were analyzed using the PEAKS Studio software (version XPro, Bioinformatics Solutions Inc., Waterloo, ON, Canada). The data were searched against the human protein database UniProtKB (http://www.uniprot.org, release 2015_10, 88500 TrEMBL and SwissProt entries containing the amino acid sequences of the test therapeutic proteins). Searches were performed with a tolerance of 15.0 ppm (precursor mass) and 0.05 Da (fragment ions) using the unspecific digest mode. Met-sulfoxide, Asn/Gln de-amidation, and N-terminal pyro-glutamylation were considered as dynamic modifications. Results were filtered at 1% false discovery rate cutoff at the peptide level. The PEAKS PTM results were exported and further analyzed using dataMAPPs, an in-house-developed R-based workflow [32], to generate heat maps.

### 2.4. HLA Binding Prediction

The binding prediction of identified unique MHC-II peptides to a donor’s HLA alleles was performed using the NetMHCIIpan-4.2 server with the recommended settings [29,33]. This specific release was trained on an extended set of HLA-DQ data [33] and claims improved sensitivity and specificity towards prediction of binding to HLA-DP and -DQ receptors. The predicted rank threshold for strong (≤1%) and weak binders (>1 to ≤5%) was used to categorize peptides. Any peptide ranked beyond these limits was classified as a non-binder. All weak and strong binders were annotated according to peptide-HLA haplotype associations. The resulting Sankey diagrams were generated using SankeyMATIC (https://sankeymatic.com/build/), and Venn diagrams were generated using InteractiVenn [34].

## 3. Results

### 3.1. Clones L243, B7/21, and SPV-L3 Exhibit Chief Specificity for HLA-DR, HLA-DP, and HLA-DQ Receptors, Respectively, While MHC-II Receptor-Precipitating Reagents with Similar Reported Specificities Differ Based on Clonality

In an initial step comparing the performance of various HLA-II-receptor immunoprecipitating reagents in the MAPPs assay, we first set up a comprehensive analytical characterization method to investigate the binding specificities of the enriched MHC-II peptides characterized overall to the HLA alleles present in a given donor. In particular, we wanted to demonstrate unequivocally an enrichment of additional HLA-DP and HLA-DQ-specific compound-derived peptides compared to a conventional workflow using an HLA-DR-only immunoprecipitating reagent. To this end, following peptide identification via the MAPPs assay using the alleged HLA-DR antibody clone L243 (our gold standard), alleged HLA-DP antibody clone B7/21, alleged HLA-DQ antibody clones SPV-L3 and 1a3, and alleged HLA-II pan antibody clones CR3/43, WR18, and Tü39, we inferred the nature of the immunoprecipitated receptors by investigating the enriched MHC-II peptides’ binding characteristics with NetMHCIIpan-4.2 [29,33]. By additionally using an HLA-II pan-specific antibody during affinity purification, we hypothesized that we might obtain a larger dataset highly enriched in HLA-DP- and HLA-DQ-specific peptides, and with possibly significantly more HLA-DR peptide ligands than using clone L243 alone.

The predicted binding specificities of the identified MHC-II peptides to the known HLA alleles specific to each investigated donor were classified as “strong binders”, weak binders”, or “non-binders”, according to the following criteria: An MHC-II peptide was designated as a “strong binder” if the predicting binding score %rank was ≤1.0 compared to a set of random natural peptides. Similarly, a weak binder was defined when the predicting binding score %rank was >1.0 to ≤5.0. A “non-binder” peptide had a predicting binding score %rank > 5.0. Thus, according to NetMHCpan-4.2, these non-binding peptides identified via LC-MS/MS were not expected to interact specifically with receptors encoded by any of the donor’s HLA alleles. Further, some MHC-II peptides were also predicted to interact with two or more alleles present in a donor; in this case, MHC-II peptides were classified as mixed interactors, wherein the binding affinity for each individual receptor should be minimally classified as “weak binder”.

The immunopeptidome data obtained from the seven considered MHC-II-precipitating antibodies was compared from three donors (Figure 1). Overall, the immunopeptidome HLA allelic relative distribution (i.e., the ratio of DR to DP to DQ MHC-II-specific peptides) was consistent for each of the antibody clones across all donors despite differing genotypes. Considering first “mono-allele specific” precipitating antibodies, peptide binders identified by the SPV-L3 and 1a3 clones appeared to be largely specific for the HLA-DQ alleles, with the latter performing significantly worse than the former. While the peptide binders identified by the B7/21 clone were largely specific for the HLA-DP alleles, for the L243 clone, 40.2–57.7% of the peptides were strictly specific for HLA-DR alleles; the majority of the remaining peptide binders identified by the L243 clone were mixed interactors, e.g., HLA-DR/DP, HLA-DR/DQ, or HLA-DR/DP/DQ binders. Interestingly, for all antibody clones, mixed interactors peptides binding to the HLA-DR/DQ alleles were more prevalent than these binding to the HLA-DR/HLA-DP alleles, whilst the occurrence of mixed interactors HLA-DP/DQ binders was almost non-existent (mean occurrence of 1.9%; stdev 0.9%).

Finally, the percentage of non-binders MHC-II peptides varied between donors. This could be attributed to differing genotypes and/or to the predictive power of the NetMHCIIpan algorithm. In our study, for example, Donor 1 and 2 expressed four distinct DRB isoforms whereas Donor 3 expressed only three distinct DRB isoforms.

The immunopeptidomes obtained with the three MHC-II pan clones closely resembled what was obtained with clone L243 with the addition of a significant portion of peptide binders strictly specific for the HLA-DP alleles (16.6% mean occurrence; stdev 1.1%). The identification of HLA-DQ strict binders was rather scarce, however, especially in comparison to clone SPV-L3 (labeled HLA-DQ-specific). Additionally, rather unexpectedly, all three MHC-II pan clones demonstrated a lower yield in HLA-DR-derived peptide binders than the L243 clone. Taken together, these data suggest that MHC-II pan antibodies may not entirely exhibit pan-specific tendencies.

We investigated this finding in more detail by comparing the immunopeptidome obtained using the HLA-DR-specific clone L243 with the dataset obtained using the MHC-II pan clone WR18, which generated the largest dataset of the three pan-HLA antibodies (Appendix A). The resulting Sankey diagrams (Figure 2) demonstrated that, while a majority of total unique peptides (40.7–53.6%, annotated as “Common”) overlapped using both reagents, 10.5–15.5% of total unique peptides strictly specific for HLA-DR alleles were identified by clone L243 only. In contrast, the strict HLA-DR-specific peptides identified by WR18 alone contributed between 3.2 and 5.6% of total unique peptides, and also contributed to almost all total HLA-DP-specific peptides. Similar results were obtained comparing the immunopeptidomes obtained with clone L243 and MHC-II pan clones CR3/43 and Tü39 (Appendix A).

We next investigated the immunopeptidomes of the two HLA-DQ-specific clones (Figure 3), as these allele’s specific peptides were not enriched by the MHC-II pan clones. The Sankey diagrams demonstrate that a large majority of HLA-DQ peptide binders were enriched using clone SPV-L3 (“Common” and SPV-L3 combined) while only a few were identified using clone 1a3. However, unexpectedly, both SPV-L3 and 1a3 exhibited relatively strong specificities for HLA-DR/DQ and HLA-DR receptors. Furthermore, clone 1a3 also exhibited some HLA-DP specificity; stronger than clone SPV-L3.

### 3.2. Application of HLA-DP, HLA-DQ, and MHC-II Pan Antibodies in the MAPPs Assay Increases the Identified Compound-Specific Peptide Repertoire, and Compound-Derived Clusters Demonstrate Differing Specificities to MHC-II Alleles

As our previous results demonstrate that the use of additional MHC-II-precipitating reagents in the MAPPs assay increases the allelic variety of the enriched peptide repertoire, we sought to determine whether this observation would hold true for the identification of compound-specific peptides. In particular, we wanted to investigate whether we could identify additional HLA-DP and HLA–DQ peptides that would not be captured if using clone L243 alone. For this exercise, MHC-II peptides derived from the fully human mAbs adalimumab and anti-interleukin (IL)-21 receptor (ATR-107) were identified and characterized according to MHC-II receptor specificity. Adalimumab, which is an anti-human tumor necrosis factor (TNF)-α antibody prescribed for the treatment of rheumatoid arthritis, psoriatic arthritis, Crohn’s disease, and other auto-immune diseases [35,36,37], is known to elicit ADA responses in 23% of patients [38]. ATR-107, which was anticipated to be used for the treatment of Crohn’s disease, was discontinued during Phase I trials because it induced ADA responses in more than 75% of subjects treated [39]. Additionally, the peptides identified from the mannose receptor C-type 1 (MRC1) protein were used to strengthen findings and depict an example of peptides derived from an endogenous protein.

The identified adalimumab, ATR-107, and MRC1-derived peptides were depicted as clusters in accompanying heatmaps via our in-house dataMAPPs strategy [32]. Clusters, which signify potential T cell epitopes, are multiple sets of unique peptides (i.e., peptides with individual/unique amino acid sequences) sharing a binding core sequence region of MHC-II receptors. A color-coded table exemplifying the binding capacities of the compound-specific unique peptides for each cluster and the resulting annotation is listed in Appendix A. Unique peptides of clusters containing post-translational modifications (PTMs), flagged by the magenta-highlighted amino acids in the heatmaps, were not included in this table since it is currently not possible to analyze peptides with PTMs in NetMHCIIpan.

An MHC-II peptide heatmap derived from the MAPPs analysis of the two antibody compounds, adalimumab and ATR-107, and the endogenous protein MRC1 is depicted in Figure 4. It is immediately apparent that the individual use of the seven immunoprecipitating reagents leads to the generation of unique cluster profiles that are reflective of the targeted HLA receptors being immune-precipitated. For example, HLA-DQ-specific clones SPV-L3 and 1a3 collectively led to the specific identification of adalimumab-specific clusters 1, 6, 7, 15, and 16 (Donor 1 and 2; Figure 4A); for ATR-107, clusters 5, 7, 10, 11, and 23 (Donor 2 and 3; Figure 4B); and for MRC1, clusters 10, 12, 30, and 41 (Donor 2 and 3; Figure 4C). Several of the clusters were not always enriched by clone 1a3, such as adalimumab-specific clusters 1, 6, and 7, and ATR-107-specific clusters 5, 7, and 10, and all clusters identified by 1a3 exhibited a lower abundance than those identified by SPV-L3; this might reflect a lower affinity of the clone 1a3 for the HLA-DQ receptors, as already noted above.

Based on the NetMHCIIpan analysis, adalimumab-derived clusters 1 and 16 (Appendix A), ATR-107-derived clusters 5, 10 (Appendix A), and 11 (Appendix A), and MRC1-derived clusters 12 and 30 (Appendix A) were most likely to be HLA-DQ receptor-restricted. In some other examples, a detailed analysis of the peptides being assigned to clusters may reveal a more complicated picture. In our analysis, a cluster is collectively made up of the peptides sharing a common binding core [32]. However, depending on the peptides’ flanking residues, the NetMHCIIpan algorithm may categorize individual MHC-II peptides of the same cluster to be restricted to different HLA alleles, which makes it in some cases difficult to precisely assign a cluster according to MHC-II annotations. For example, the individual peptides part of the adalimumab-derived cluster 15 (Donor 2; Figure 4A, Appendix A) and of the identical ATR-107 cluster 18 (Donor 2; Figure 4B, Appendix A) were assigned binding specificity for either HLA-DQ or HLA-DR/DQ alleles. However, in the case of the peptides being categorized as mixed HLA-DR/DQ interactors, some were strong binders for HLA-DQ alleles and weak binders for HLA-DR alleles, while others were weak binders to both HLA-DR and HLA-DQ alleles. In this context, it is telling that, in Donor 2, all peptides of these clusters were exclusively captured using the HLA-DQ-specific clones SPV-L3 and 1a3 while there was no signal using the HLA-DR-specific L243 clone or the pan-MHC-II clones. In contrast, in Donor 1, one peptide of the ATR-107 cluster 18 was captured using the HLA-DR specific clone L243 (assigned DR/DQ; Appendix A) while there was no signal using the HLA-DQ-specific clones SPV-L3 and 1a3. These findings demonstrates that blanket assignment of MHC-II peptide clusters to specific alleles may not be possible based on a limited number of donors, as shown here.

For the HLA-DP-specific B7/21 clone, only the adalimumab-specific cluster 17 (Donor 3; Figure 4A, Appendix A), corresponding to the identical ATR-107-specific cluster 24 (Donors 1–3; Figure 4B, Appendix A) were unique to this clone; this cluster was localized at the C-terminus of the Fc fragment and was detected at rather low abundance, except for Donor 2. Interestingly, all peptide parts of this cluster were considered as “non-binders” to all the donors’ specific alleles. No MRC1-specific clusters were unique to the B7/21 clone.

Since we previously observed that the MHC-II pan antibodies were not as pan-specific as originally presumed (e.g., low HLA-DQ receptor specificity), it was not surprising that their compound-specific peptide repertoire did not include many of the clusters enriched by the SPV-L3 clone (Figure 4). However, in the case of the endogenous MRC1-derived peptides, almost all of the clusters identified by the HLA-DR-, HLA-DP-, and HLA-DQ-specific clones were also observed by at least one or all of the MHC-II pan antibody clones, with WR18 exhibiting the highest representation of the “mono-allele specific” precipitating antibodies (Figure 4C).

Moreover, despite the larger MHC-II peptide repertoire previously observed for clone CR3/43 than Tü39, no adalimumab-, ATR-107-, or MRC1-specific clusters were unique to clone CR3/43. In contrast, both alleged MHC-II pan-specific clones Tü39 and WR18 led to the identification of clusters that were unique to the individual clones only.

MHC-II peptide heatmaps generated using the HLA-DR-specific L243 clone were generally rich in data that were mostly (but not entirely) recapitulated using the pan-MHC-II clone WR18. ATR-107-derived cluster 1 (weak HLA-DR binders; Appendix A) and adalimumab-derived clusters 11 (strong HLA-DR binders; Appendix A) and 13 (“non-binder”; Appendix A) were identifiable by clone L243 only.

The most striking observation in this experiment was ATR-107-derived peptide clusters 13 and 14 (Figure 4B) as they were identified by all antibody clones for all donors tested. Although the abundance of this cluster varied depending on the clones that were used, the highest signal was consistently detected when using the HLA-DP-specific B7/21 and MHC II pan clones. Using NetMHCpan, for Donors 1 and 3, almost all peptides of these clusters were predicted to be specific strictly for the HLA-DP receptors (Appendix A). For Donor 2, peptides of this cluster were annotated as exhibiting strict strong HLA-DP binders, mixed HLA-DR/DP, or mixed HLA-DP/DQ receptor specificities (Appendix A).

Taking all the results together, we conclude that using multiple MHC II receptor-precipitating reagents in the MAPPs assay lead to the additional identification of compound-derived MHC II receptor-specific peptides in comparison to L243 alone. However, none of the evaluated reagents was able to individually map all the observed clusters.

### 3.3. Mixed Immunoprecipitation Strategies Lead to More Robust Compound-Specific Peptide Detection Than MHC II Antibodies Alone

Our previous experiments that characterized the specificity of seven MHC II-precipitating reagents revealed that no individual antibody clone is able to recover the complete human leukocyte antigen II peptide repertoire. Based on these results, we selected clones L243, B7/21, SPV-L3, WR18, and Tü39 to assess the potential advantages of a mixed IP strategy in the MAPPs assay in the same three previously described donors. These clones were selected based on their characterization according to specificity and the identification of compound-specific clusters that were unique to these individual clones only.

A mixed IP strategy (e.g., the IP step is performed using three or more clones concurrently), in our opinion, would make efficient use of the available lysate and of the analytical LC-MS capacity as opposed to parallel IP strategies, which consume a fresh lysate for each MAPPs assay, therefore requiring several-fold more starting cells and multiple LC-MS measurements for each sample. Additionally, a mixed IP strategy requires the IPs to be performed only once in comparison to serial IP strategies where each flow-through is used for subsequent IPs [40]. Thus, risks of MHC-II peptide–receptor complexes loss in the serial sample-handling and dilution is minimized.

We performed four mixed IP strategies, containing minimally the HLA-DR clone L243 and the HLA-DQ-specific clones SPV-L3, then adding either the anti-DP-specific-clone B7/21, the pan-MHC-II clone WR18 or the pan-MHC-II clone Tü39. We hypothesized that the use of the pan-MHC-II clones in the mixture would serve as the HLA-DP component and also lead to the identification of additional HLA-DR-specific peptides that would otherwise not be seen with the HLA-DR clone L243. Finally, we also evaluated a mixed IP containing all five IP reagents.

Across all mixed IP strategies tested, the general MHC-II peptide repertoire distribution was rather consistent (Figure 5). Taking all donors into account, each strategy led to the identification of about 30–54% HLA-DR, 3–11% HLA-DP, and 2–6% HLA-DQ-specific peptides. The majority of remaining peptides (about 20–30%) were predicted to be mixed HLA-DR/DQ binder peptides, whereas the minority were mixed HLA-DP/DQ binder peptides (about 0–1%). The percentage of “non-binders” MHC-II peptides relative to the total peptide count for the mixed IP strategies was comparable to what was observed from individual IP strategies alone (about 8–13%).

While general MHC-II peptide repertoires were rather consistent between IP strategies, the compound-specific cluster profiles (Figure 6) differed depending on the IP strategies. While desirable, it was apparent that none of the mixed IP strategies (including the one performed with five IP reagents) were able to recapitulate completely a representative compound-specific MHC-II peptide repertoire from the original cluster profiles enriched by all individual clones as previously observed for Figure 4 (designated as a mixed simulation (“mix sim”) in Figure 6). In fact, in some cases, the use of a mixed IP strategy using concurrently all five antibody clones (L243, B7/21, Tü39, WR18, and SPV-L3) resulted in a loss of signals, possibly due to competitive binding between the five antibody clones. Additionally, the combination L243, Tü39, and SPV-L3 as well as L243, WR18, and SPV-L3 IP mixes appeared to be representative of most compound-specific MHC-II peptide clusters identified from L243 alone, but with some differences, despite these including the L243 clone as well.

Not all adalimumab-derived clusters were enriched with the L243, B7/21, and SPV-L3 mix compared to the L243 and SPV-L3 mix with either WR18 or Tü39 clones. For Donor 1, adalimumab clusters localized in the VH (cluster 2) and VL (cluster 8) domains were unique to the L243 and SPV-L3 mix with either WR18 or Tü39 clones, respectively. Furthermore, the L243, WR18, SPV-L3, and L243, B7/21, SPV-L3 mix for Donor 2 and all mixed IP strategies for Donor 1 led to the identification of additional ATR-107-derived peptide clusters 6 and 8, respectively, not previously discerned when comparing the individual cluster profiles for each of the clones. The ATR-107-derived HLA-DQ-specific clusters 5 and 18 identified by SPV-L3 alone in Figure 6 were both identifiable in the L243, SPV-L3 mix with either WR18 or Tü39 clones in Figure 4.

Donor 1 was the only circumstance in which the L243, B7/21, and SPV-L3, as well as the L243, B7/21, Tü39, WR18, and SPV-L3 mixes each led to the identification of an additional ATR-107-derived cluster localized in the VH domain (cluster 3 and 1, respectively). This is surprising because the cluster identified with the L243, B7/21, Tü39, WR18, and SPVL-3 mix strategy was also not previously observed for this donor.

Next, we compared the specificities (i.e., whether a peptide binder is specific for, e.g., HLA-DR only or a mixed reactor) of the identified compound-derived MHC-II peptides for L243 alone versus the mixed IP strategies for each donor (Appendix A). In this way, we could determine the number of peptides that were unique to a certain IP strategy, as well as the number of peptides that overlapped across multiple IP strategies. Our comparisons of the overlapping compound-derived MHC-II peptide repertoire identified for L243 versus the mixed IP strategies revealed that clone L243 consistently led to the majority of HLA-DR-specific peptides across all donors. As donor variability (i.e., the HLA alleles present in a given donor) and/or the compound may impact the peptide repertoire, the mixed IP strategies and clone L243 alone were ranked according to the number of compound-specific peptides and predicted binding specificities (Table 1). This simplified ranking strategy was performed in an empirical way, in which the strategy that led to the identification of the highest number of compound-specific peptides (per binding specificity (“annotation”) and donor) corresponded to the highest ranking on a scale of 1 to 5, where 1 is the best (highest number of compound-specific peptides identified) and 5 is the worst (lowest number of compound-specific peptides identified). For example, for Donor 1, clone L243 alone led to the highest number of peptides that were predicted to bind to HLA-DR receptors only (27 peptides; ranked 1), followed by the mixed IP strategies of L243, Tü39, SPV-L3 (24 peptides; ranked 2), L243, B7/21, SPV-L3 (23 peptides; ranked 3), L243, WR18, SPV-L3 and L243, B7/21, Tü39, WR18, SPV-L3 (both 22 peptides; both ranked 4).

The ranking was implemented across all donors to determine the most optimal strategy for potential T cell epitope identification. The ranks of each IP strategy across all donors and peptide annotations were then summed. Overall, the mixed IP strategy of L243, WR18, and SPV-L3 exhibited the highest ranking (total 35; rank (1)), corresponding to the overall highest number of compound-specific peptides identified, consecutively followed by (2) L243, B7/21, SPV-L3 (total 39); (3) clone L243 alone (total 40); (4) L243, B7/21, Tü39, WR18, and SPV-L3 (total 45); and (5) L243, Tü39, SPV-L3 (total 51).

Taking together the compound-derived MHC-II peptide cluster profiles (Figure 6), overlap of the compound-specific peptide repertoires (Appendix A), and ranking of the mixed IP strategies (Table 1) in comparison to clone L243 alone, using L243 alone is still advantageous to recapitulate a majority of compound-specific peptide clusters. If interested in the additional identification of HLA-DP and HLA-DQ-specific peptides, our ranking suggests that a mixed IP strategy with clones L243, WR18, and SPV-L3 is most optimal.

Overall, although MHC-II receptor-precipitating reagents with similar reported specificities differ based on the clone used, our characterization and specificity analysis of the various HLA-DP, HLA-DQ, and MHC-II pan antibody clones in the MAPPs assay reveals that the utility of these antibodies leads to an augmented identification of HLA-DP and HLA-DQ receptor-specific peptides.

## 4. Discussion

Researchers have suggested that a higher probability of epitopes recognized by CD4^+^ T cells are HLA-DR receptor-restricted in contrast to HLA-DP and HLA-DQ receptors. This claim can be attributed to the large overlap between DRB1 and DRB3/4/5 loci, whereas HLA-DP exhibits an intermediate pattern of repertoire overlap, and HLA-DQ is associated with a mostly unique repertoire [15]. In addition, studies have shown that HLA-DR receptors are often found to bind with strong affinity to mAb-specific T cell epitopes [41,42,43]. For example, the CDR3 of the heavy chain of the fully human mAb adalimumab is a hotspot of strong peptide binders to HLA-DR receptors, and consequently, this region contains the vast majority of identified T cell epitopes of adalimumab [42,43].

Due to the large data available for HLA-DR-specific binding predictions, the majority of prediction algorithms as well as MAPPs studies have currently been restricted to identifying HLA-DR-specific epitope candidates. The latter may also be attributed to the absence of well-characterized HLA-DP, HLA-DQ, or MHC-II pan antibodies with appropriate receptor-binding performance available for the MAPPs assay, which results in a limited yield in the immunoprecipitation step [21,33]. As such, data available regarding adalimumab-specific epitopes restricted for HLA-DP and HLA-DQ are limited, and this represents an important gap of knowledge.

Recently, the limited predictability of HLA-II epitope-prediction algorithms in the context of HLA-DQ receptors was greatly improved with the enhanced NetMHCIIpan-4.2 version, claiming an equal predictive performance for the associated HLA-DR and HLA-DQ datasets [33]. In this study, we devised a MAPPs–NetMHCIIpan coupled methodology, which allowed us to generate high-quality data to gain more insight into the specificities of identified peptides binding to MHC-II receptors. Using this methodology, we describe an approach to accommodate MHC-II pan receptors for improved predictability of potential T cell epitopes in the MAPPs assay.

This study consisted of testing various clones of MHC-II pan (CR3/43, WR18, and Tü39), HLA-DP (B7/21), and HLA-DQ (SPV-L3 and 1a3) receptor-immunoprecipitating reagents and comparing them with the well-known and well-performing HLA-DR antibody clone L243 (our gold standard). These clones were tested as individual IP preparations followed by various mixed IP strategies. Endogenous and compound-specific peptides per antibody clone were identified via the MAPPs assay, and then further analyzed by NetMHCIIpan-4.2 to predict the binding capacity of the peptides to any HLA-II alleles of known sequence, and to ascertain whether the various MHC-II antibody clones and mixed IP strategies exhibit specificity for HLA-DR, HLA-DP, and/or HLA-DQ receptors.

Based on the compound-specific peptide cluster profiles for each of the individual clones, we deduced that the application of HLA-DP, HLA-DQ, and MHC-II pan antibodies in the MAPPs assay increases the identified MHC-II peptide repertoire and the compound-specific cluster profile. A lower number of total unique peptides as well as compound-specific clusters were discernible for the HLA-DP and HLA-DQ receptor-specific clones compared to HLA-DR and MHC-II pan clones. These data are consistent with studies that reported significantly weaker responses and lower number of epitopes for HLA-DP and HLA-DQ loci combined when compared to the HLA-DR locus [15,44,45,46].

We observed that the alleged MHC-II specificities of the antibody clones differ from manufacturing claims. For example, MHC-II pan antibodies do not entirely exhibit pan-specific tendencies. These antibody clones do not capture all HLA-DR receptors compared to clone L243 and exhibit a weak affinity to HLA-DQ receptors.

Furthermore, although the investigated “mono-allele specific” MHC-II antibody clones are claimed to be solely specific for a certain receptor, we observed the occurrence of other MHC-II receptor-derived peptides than the HLA receptor of interest. For example, even though clone 1a3 is allegedly specific for HLA-DQ receptors, our analysis revealed that this clone simultaneously leads to the identification of HLA-DP restricted peptides (Appendix A). These so-called “co-immunoprecipitated contaminants” are generally found in MS-eluted ligand datasets, despite the use of receptor-specific antibodies, due to the function of the detergent used during cell lysis. This results in a piece of membrane that carries the MHC-II receptor of interest in addition to contaminating/non-specific MHC-II receptors [47,48]. To circumvent this, researchers suggest the use of engineered cells that secrete the soluble form of MHC-II receptors [49,50]. However, this strategy is not possible when exploiting biological or tissue samples [47].

For all antibody clones and various mixed IP strategies, we observed that peptides demonstrate binding capacities to multiple receptors, which hinders the annotation of peptides as distinct motifs. A large fraction of peptides were consistently HLA-DR/DQ-specific, an intermediate fraction was associated with HLA-DR/DP or HLA-DR/DP/DQ, and if present, an extremely minor fraction was HLA-DP/DQ-specific. Marcu et al. [51] also reported that a single peptide sequence can be a binder against multiple haplotypes of the same donor. The results of Grifoni et al. [15] suggest that the epitope repertoire overlaps across loci, which could be based on similarities in the peptide-binding motifs of alleles from an individual [15,52].

The rare occurrence of HLA-DP/DQ-specific peptides could possibly be attributed to the structural properties of both these receptors. Since the α and β chains are polymorphic, the likelihood that a peptide binds to one of the four different HLA-DP combinations as well as one of the four different HLA-DQ combinations is minimal [53].

The presence of non-binders against any subject’s haplotype seems to be a common phenomenon [40,51], albeit with differing propensities depending on the antibody clone. For example, between 19–50% of total unique peptides were considered “non-binders” when using either of the HLA-DQ-specific clones SPV-L3 and 1a3. In contrast, the presence of non-binders for the remaining individual clones and all of the mixed IP strategies was between 7–19% and 8-13%, respectively, which correlates with the predictive performance of NetMHCIIpan. According to the Immune Epitope Database (IEDB), ref. [54] NetMHCIIpan-4.0 has an overall 72% performance ranking, and Nilsson et al. [33] asserts that version 4.2 exhibits a significant gain in predictive performance. In this regard, it is worth mentioning that NetMHCIIpan-4.2 is able to predict the binding capacity of low-abundance peptide clusters, such as the adalimumab- and ATR-107-derived clusters 15 and 18, respectively, which were identified by the clones SPV-L3 and 1a3 (Figure 4) and predicted to be strong binders to HLA-DQ or mixed HLA-DR/DQ interactors (Appendix A).

Unexpectedly, some compound-specific peptide clusters overlapping CDR regions were predicted to be non-binders, such as ATR-107-derived cluster 12, which fully and/or partially overlaps the CDR3 region of the VL domain. Harding et al. [55] concludes that CD4^+^ T cell epitopes occur only in CDR-containing regions of human(ized) antibody V regions due to the introduction of somatic hypermutations (i.e., amino acid exchanges) during the affinity maturation process of mAb development. It is therefore surprising that all unique peptides of the fully human ATR-107-derived cluster 12 were predicted to be non-binding peptides. It is unlikely that this cluster is a contaminant due to its high abundance across all L243, most MHC-II pan, and all mixed IP preparations for all donors. With hesitancy, we are inclined to attribute the annotation of this non-binder cluster to the limited performance of epitope prediction algorithms. This peptide cluster might not be accurately predicted by binding affinity algorithms.

Due to the polymorphisms in the MHC-II genes [5,56,57], the immunodominant CD4^+^ T cell epitopes of mAbs can significantly differ between individuals as a function of their HLA genotype [42,58,59,60], but can also be shared by multiple donors as a result of common binding specificities of MHC-II receptors [42,61]. For example, this explains why our adalimumab cluster 4, which was identified by Meunier et al. [43] in the MAPPs assay and regarded as a potential T cell epitope, is identifiable in samples precipitated with HLA-DR-specific clone L243 and all MHC-II pan clones (CR3/43, WR18, and Tü39) for Donors 1 and 3, whose HLA genotypes differ.

Surprisingly, compound-derived unique peptides of a given cluster demonstrated differing specificities to MHC-II alleles despite sharing a peptide binding core. This suggests that not only the binding core, but the peptide-flanking residues also influence the binding capacity to MHC-II receptors. This finding was confirmed by several studies, in which the peptide length as well as the peptide-flanking residues (in particular the residue at position P-1) significantly influence the affinity for MHC class II molecules and T cell recognition [62,63,64,65].

Nilsson et al. [33] suggests that the large performance gap between HLA-DR versus HLA-DQ (and HLA-DP) can be to a very large degree due to earlier immunoprecipitation studies [66] that first depleted for HLA-DR receptors followed by using a pan-HLA class II antibody to enrich for HLA-DQ and HLA-DP data. This generally led to lower quantity and quality of ligands obtained in HLA-DQ studies. Other studies indicated that serial IP strategies often risk contamination of the second immunoprecipitation by the first [40]. Owing to these technicalities, our strategy of applying a minimum of three antibody clones would be unpractical in a serial IP.

Thus, we selected the four top-performing antibody clones (B7/21, SPV-L3, WR18, and Tü39) in terms of specificity, MHC-II peptide repertoire, and compound-specific cluster profile to perform various mixed IP strategies in comparison to using clone L243 alone. In some instances, the mixed IP with all five antibody clones (L243, B7/21, SPV-L3, WR18, and Tü39) exhibited limited efficiency in terms of compound-specific peptide enrichment in contrast to the mixed IPs with MHC-II pan clones WR18 (L243, WR18, SPV-L3) or Tü39 (L243, Tü39, SPV-L3). This may be due to a capacity issue of the streptavidin beads in the affinity chromatography cartridges when using five antibody clones at once, and therefore, the saturation to streptavidin beads need to be further explored.

The use of a pan-HLA antibody is in line with previous HLA II peptidome studies in which leveraging a mixture of HLA-DR (clone L243) and MHC-II pan (clone Tü39) receptor-specific antibodies enabled more efficient MHC-II peptide purification [51,53]. In our study, we additionally included the HLA-DQ-specific clone SPV-L3 in our mixed IP strategies since our preliminary results revealed that the MHC-II pan clones tested exhibit weak specificities for HLA-DQ receptors. Applying a mixed IP strategy that includes the HLA-DR-specific clone L243, HLA-DQ-specific clone SPV-L3, as well as the MHC-II pan clone WR18 in the MAPPs assay leads to an augmented compound-specific peptide repertoire. This mixed IP strategy leads to additional compound-derived HLA-DP and HLA-DQ receptor-specific peptide binders in contrast to clone L243 alone. However, this study is based on three donors and needs to be extended to strengthen findings. Further, the quality control should be monitored over time, and optimizations of the antibody clone concentrations in the mixed IP strategy should be further explored to determine which antibody concentration ratio leads to the best MHC II immunopeptidome coverage.

Current limitations to layering MAPPs analyses and *in silico* prediction algorithms include the performance gap in peptide binding predictions between HLA-DR and HLA-DP. Although the novel data of Nilsson et al. [33] demonstrated that the predictive performance of the HLA-DR and HLA-DQ ligand datasets are equal for NetMHCIIpan-4.2, the authors note that a similar type of analysis of affinity purification combined with refined data mining and motif deconvolution should be extended to HLA-DP ligands.

Another caveat is that MAPPs-identified peptides containing PTMs, which are annotated as mass shifts of an amino acid within the peptide sequence, cannot be comprehensively considered in NetMHCIIpan-4.2. Currently, the only options to circumvent PTMs in NetMHCIIpan-4.2 are to either analyze the unmodified peptide sequence or exchange the modified amino acid with an “X”, signifying an unknown amino acid. This may pose limitations to MAPPs analyses as compound-derived peptides containing PTMs occurred in our study, usually oxidation at methionine or de-amidation of asparagine and glutamine, and researchers are highlighting the importance of epitope PTMs in immunopeptidomes. Notably, Katayama et al. [67] discovered that MHC-II-bound citrullinated peptides are a source of immunogenicity, and León-Letelier et al. [68] asserts that PTMs can induce immunogenicity more than their unmodified counterparts.

In this study, we also noticed limitations of current capture reagents for MHC-II peptidomics, such as increased ambiguity in peptide assignment and binding motif determination. Evaluations of antibody purifications from monoallelic HLA-II cell lines could address this bias [53,69,70], and also be used to determine allele-specific HLA-II-binding registers for the improvement of multiallelic deconvolution methods [53,69]. However, engineering monoallelic cell lines may introduce bias by altering processing and presentation machinery [53]. Thus, we still believe that using a multiallelic approach in MAPPs and NetMHCIIpan analyses for our intents and purposes deepens our understanding of MHC-II processing and presentation rules. As such, the future perspective of generating an in-house MHC-II pan antibody remains worthwhile.

Lastly, if costs are not to be considered (reagent costs per sample were in the low 2-digit figure for L243 alone, and almost 50× higher for the L243, WR18, and SPV-L3 combination), the results reveal the advantage of incorporating a mixture of MHC-II receptor-precipitating reagents into the MAPPs assay due to the increased identified repertoire of HLA-DP and HLA-DQ compound-derived peptides. However, utilizing clone L243 alone is best and extremely cost-effective if only interested in compound-derived HLA-DR-specific peptides.

In future, the implementation of our optimized HLA-DR, HLA-DP, HLA-DQ receptor-specific mixed IP strategy into a novel MAPPs assay-based personalized healthcare tool has the potential to be an invaluable clinical tool to predict the immunogenic potential of therapeutic mAbs in patients prior to treatment.

## 5. Conclusions

In this study, we sought to expand the MAPPs assay to accommodate MHC-II receptors for improved predictability of T cell epitopes during preclinical immunogenicity risk assessments. Due to the lack of satisfactory MHC-II pan, HLA-DP, and HLA-DQ receptor-specific antibodies available, the MAPPs assay typically exploits the well-performing (in terms of binding capacity) HLA-DR antibody (clone L243; our gold standard). Therefore, we compared and characterized various MHC-II-precipitating reagents (MHC-II pan clones CR3/43, WR18, and Tü39; HLA-DP clone B7/21; HLA-DQ clones SPV-L3 and 1a3) according to their specificity. Peptides identified in the MAPPs assay were analyzed via the HLA-epitope prediction algorithm NetMHCIIpan-4.2 to define their binding capacities. These reagents were assessed in individual IP preparations as well as mixed IP strategies consisting of a minimum of three clones with differing specificities.

Although using clone L243 is still best and cost-effective if interested solely in compound-derived HLA-DR-specific peptides, we recommend using a mixed IP strategy of L243, SPV-L3, and MHC-II pan clone WR18 for the additional identification and characterization of a broader range of HLA-DP and HLA-DQ-specific T cell epitopes. Utilizing this mixed IP strategy to layer MAPPs and NetMHCIIpan analyses is vital to identify specific alleles of individuals at risk in response to therapeutic mAb treatment. The respective peptide repertoires will provide insights into DC peptide processing and presentation rules that govern T cell responses. Taken together, layering our MAPPs and NetMHCIIpan analysis strategy will aid in the identification of MHC-II alleles and their presented peptides, which are a source of potential immunotherapeutic targets and disease biomarkers.

## Figures and Tables

**Figure 1 biology-12-01265-f001:**
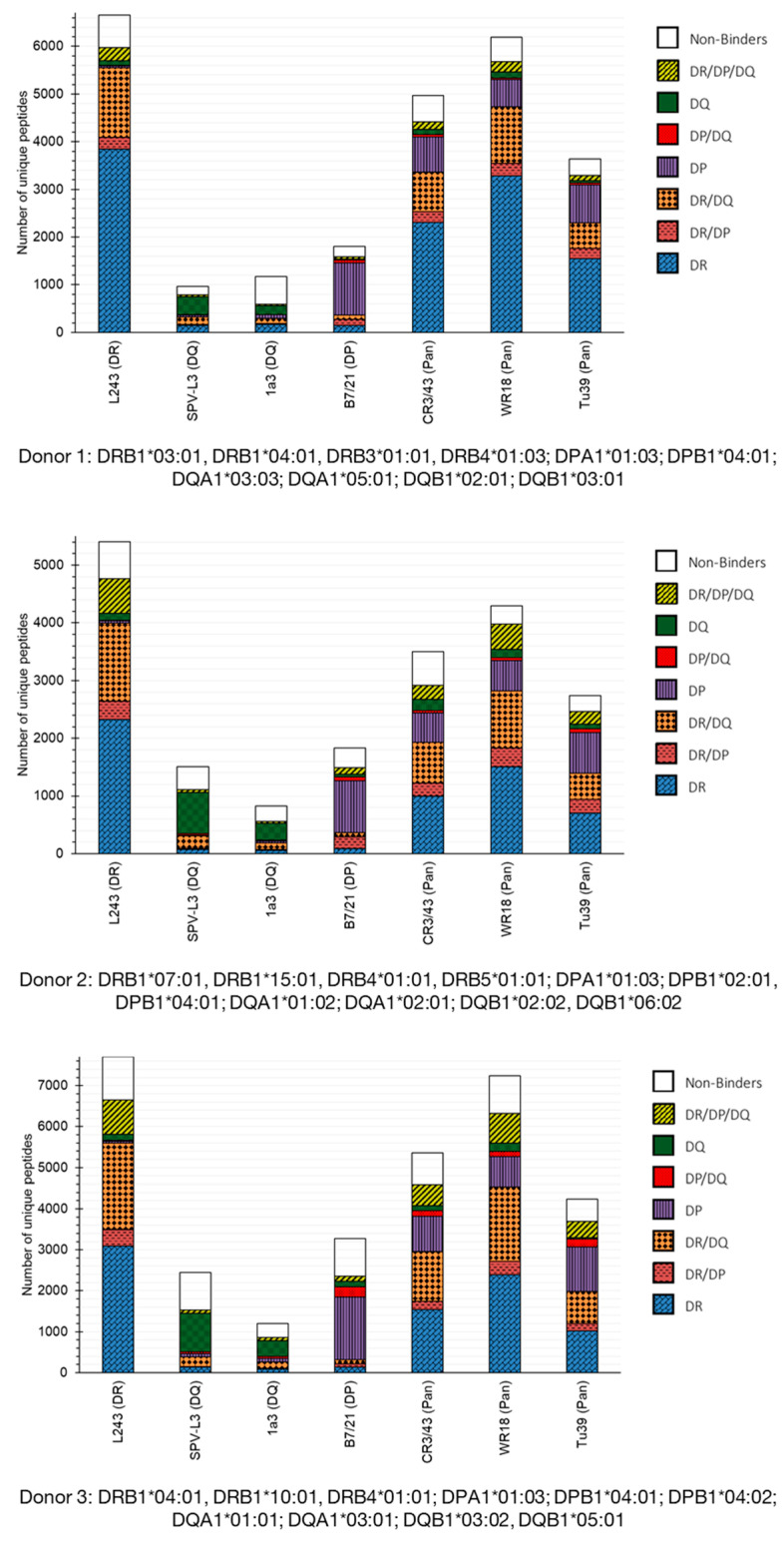
**Immunopeptidome profiles for various major histocompatibility complex (MHC) II receptor-precipitating antibodies.** Alleged anti-human leukocyte antigen (HLA)-DR clone L243, anti-HLA-DQ-specific clones SPV-L3 and 1a3, anti-HLA-DP-specific clone B7/21, and anti-MHC-II pan clones CR3/43, WR18, and Tü39 were characterized according to MHC-II receptor specificities. MHC-II-associated peptide proteomics (MAPPs) assay-identified peptides were analyzed via NetMHCIIpan to predict the binding capacities of peptides to the HLA alleles specific for each donor tested. Unique peptide binders are annotated according to binding capacity (i.e., binding to HLA-DR, HLA-DP, HLA-DQ only, or multiple alleles (HLA-DR/DP, HLA-DR/DQ, HLA-DP/DQ, and HLA-DR/DP/DQ)). Shown are the total unique MHC-II peptide repertoires identified for each of the antibody clones for three donors. The genotype for each donor is included below the respective graph.

**Figure 2 biology-12-01265-f002:**
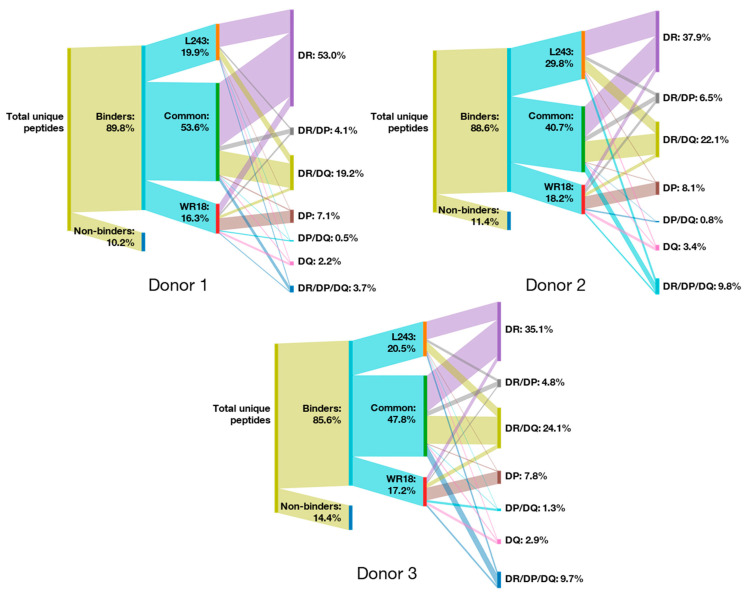
**Comparison of major histocompatibility complex (MHC) II peptide repertoires between human leukocyte antigen (HLA)-DR receptor-precipitating antibody clone L243 versus MHC-II pan-precipitating antibody clone WR18**. Sankey diagrams for each donor depicting the proportional peptide distribution between MHC-II receptor-specific antibody clones WR18 and L243. The height of the branches is proportional to the distribution. Shown are the percentage of binders and non-binders with respect to the total unique peptide count, and the percentage of peptides identified by clone L243 only and clone WR18 only. “Common” relates to the overlapping unique peptides (i.e., peptides identified by both clones). The distribution of specific MHC-II receptor binders is representative of the total number of MHC-II receptor-specific peptides, and the percent distribution corresponds to the total unique peptide count.

**Figure 3 biology-12-01265-f003:**
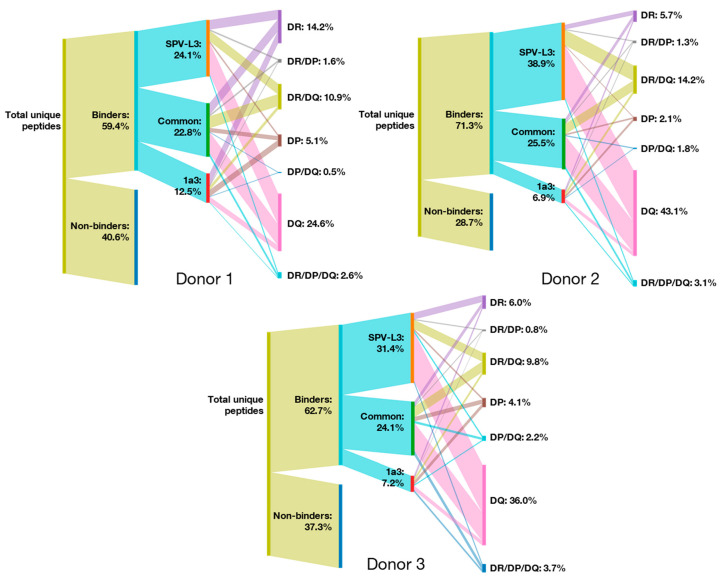
**Comparison of major histocompatibility complex (MHC) II peptide repertoires between alleged human leukocyte antigen (HLA)-DQ receptor-precipitating antibody clones SPV-L3 versus 1a3**. Sankey diagrams for each donor depicting the proportional peptide distribution between HLA-DQ receptor-specific antibody clones SPV-L3 versus 1a3. The height of the branches is proportional to the distribution. Shown are the percentage of binders and non-binders with respect to the total unique peptide count, and the percentage of peptides identified by clone SPV-L3 only and clone 1a3 only. “Common” relates to the overlapping unique peptides (i.e., peptides identified by both clones). The distribution of specific MHC-II receptor binders is representative of the total number of MHC-II receptor-specific peptides, and the percent distribution corresponds to the total unique peptide count.

**Figure 4 biology-12-01265-f004:**
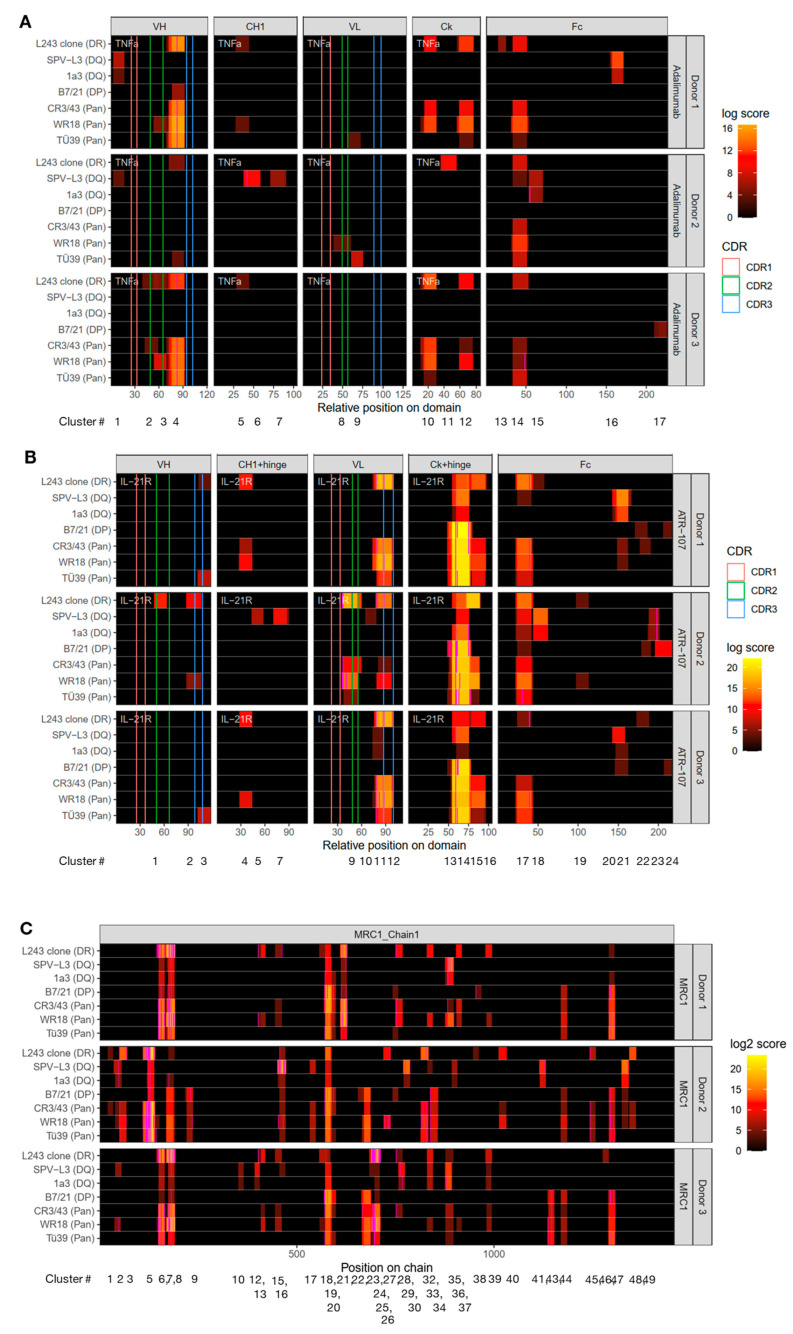
**Additional application of human leukocyte antigen (HLA)-DP, HLA-DQ, and pan-HLA antibodies in the major histocompatibility complex (MHC)-II-associated peptide proteomics (MAPPs) assay increases the identified compound- and MRC1-specific peptide repertoire in comparison to anti-HLA-DR clone L243 alone**. Heatmaps depicting the cluster profile of MAPPs-identified (**A**) adalimumab, (**B**) ATR-107, and (**C**) MRC1-specific peptides respective to MHC-II-precipitating antibody clones L243, SPV-L3, 1a3, B7/21, CR3/43, WR18, and Tü39. The adalimumab and ATR-107 sequence regions are organized according to the antibody domains (i.e., variable domain of the heavy chain (VH), constant domain of the heavy chain (CH1), variable domain of the light chain (VL), constant region of the kappa-type light chain (Ck), and the fragment crystallizable (Fc) region). Vertical pink, green, and blue lines along the sequence of the VH and VL domains correspond to the position of the complementarity-determining regions (CDRs) 1 to 3. Identified peptide clusters, annotated as C1 to C17 for adalimumab, C1 to C24 for ATR-107, and C1 to C49 for MRC1, are depicted as colored regions with varying abundances (as a log score) per sequence position, spanning from dark red to yellow. Donor number is denoted on the vertical axis, and the MHC-II antibody clones tested for each donor signify the individual horizontal bars of the heatmap. Clusters are consistently numbered across all heatmaps in this study (see Figure 5). All clusters and peptide sequences are listed in Appendix A.

**Figure 5 biology-12-01265-f005:**
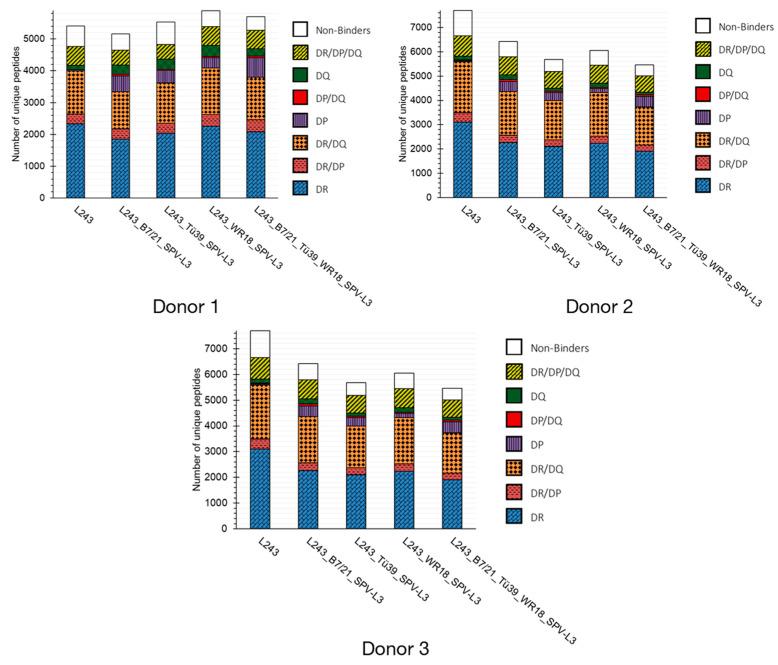
**Immunopeptidome profiles for mixed immunoprecipitation (IP) strategies using various combinations of major histocompatibility complex (MHC) II receptor-precipitating antibodies with differing specificities**. (Furthest left) Clone L243 alone as a gold standard reference, and mixed IP strategies performed with (far left) L243, B7/21, and SPV-L3 clones; (left) L243, Tü39, and SPV-L3 clones; (right) L243, WR18, and SPV-L3 clones; and (far right) L243, B7/21, Tü39, WR18, and SPV-L3 clones. MHC-II-associated peptide proteomics (MAPPs) assay-identified peptides were analyzed via NetMHCIIpan to predict the binding capacities of peptides to the HLA alleles specific for each donor tested. Unique peptide binders are annotated according to binding capacity (i.e., binding to HLA-DR, HLA-DP, and HLA-DQ only, or multiple alleles (HLA-DR/DP, HLA-DR/DQ, HLA-DP/DQ, and HLA-DR/DP/DQ)). Shown are the total unique MHC II peptide repertoires identified for three donors.

**Figure 6 biology-12-01265-f006:**
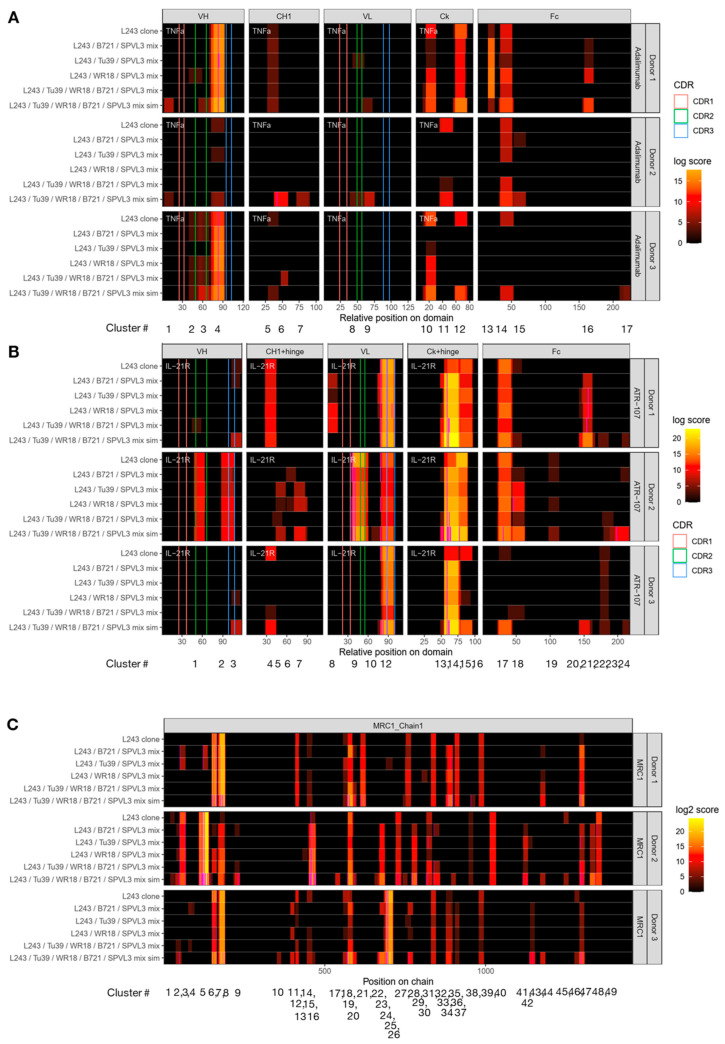
**Identification of peptide cluster profiles when applying various mixed immunoprecipitation (IP) strategies to the major histocompatibility complex (MHC) II-associated peptide proteomics (MAPPs) assay**. Heatmaps depicting the cluster profile of MAPPs-identified (**A**) adalimumab, (**B**) ATR-107, and (**C**) MRC1-specific peptides respective to various IP mixtures of MHC-II-precipitating antibody clones L243, SPV-L3, and either B7/21, WR18, or Tü39, the combination of all five antibody clones, and a simulated mixture depicted from the individual clones run in a single IP. The adalimumab and ATR-107 sequence regions are organized according to the antibody domains (i.e., variable domain of the heavy chain (VH), constant domain of the heavy chain (CH1), variable domain of the light chain (VL), constant region of the kappa-type light chain (Ck), and the fragment crystallizable (Fc) region). Vertical pink, green, and blue lines along the sequence of the VH and VL domains correspond to the position of the complementarity-determining regions (CDRs) 1 to 3. Identified peptide clusters, annotated as C1 to C17 for adalimumab, C1 to C24 for ATR-107, and C1 to C49 for MRC1 are depicted as colored regions with varying abundances (as a log score) per sequence position, spanning from dark red to yellow. Donor number is denoted on the vertical axis, and the MHC-II antibody clones tested for each donor signify the individual horizontal bars of the heatmap. Clusters are consistently numbered across all heatmaps in this study (see Figure 4). All clusters and peptide sequences are listed in Appendix A.

**Table 1 biology-12-01265-t001:** Ranking of the mixed immunoprecipitation (IP) strategies and human leukocyte antigen (HLA)-DR-specific clone L243 alone. The various mixed IP strategies and anti-HLA-DR clone L243 alone were ranked based on the compound-specific peptide repertoires and predicted MHC-II receptor binding specificities (annotated as DR, DP, DQ, DR/DQ, DR/DP, DR/DP/DQ, and non-binders). The figures depicting the overlap of the identified peptides per strategy for each donor (Appendix A) were used for ranking. Ranking on a 1 to 5 scale where 1 is best (highest number of compound-specific peptides identified) and 5 is worst (lowest number of compound-specific peptides identified). The overall rank of each IP strategy is indicated.

Donor	Peptide Annotation	L243	L243_B7/21_SPV-L3	L243_Tü39_SPV-L3	L243_WR18_SPV-L3	L243_B7/21_Tü39_WR18_SPV-L3
1	DR	1	3	2	4	4
1	DP	5	2	4	3	1
1	DQ	3	1	2	1	3
1	DR/DQ	1	2	3	3	2
1	DR/DP	0	0	0	0	0
1	DR/DP/DQ	1	2	1	1	1
1	Non-Binders	3	2	4	1	5
2	DR	1	3	3	2	4
2	DP	5	2	4	3	1
2	DQ	2	3	2	1	2
2	DR/DQ	2	2	4	1	3
2	DR/DP	5	2	4	3	1
2	DR/DP/DQ	0	0	0	0	0
2	Non-Binders	3	2	3	1	4
3	DR	1	3	2	2	4
3	DP	4	1	2	3	1
3	DQ	0	0	0	0	0
3	DR/DQ	1	2	4	2	3
3	DR/DP	0	0	0	0	0
3	DR/DP/DQ	1	4	3	2	3
3	Non-Binders	1	3	4	2	3
	Total	40	39	51	35	45
	Overall Rank	3	2	5	1	4

## Data Availability

The raw and processed mass spectrometric data have been deposited to the PRIDE archive (http://massive.ucsd.edu/ProteoSAFe/QueryPXD?id=PXD044851) via theMassIVE partner repository with the data set identifier PXD044851 (MassIVE: MSV000092747).

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
