# Peer review of "Expanding the MAPPs Assay to Accommodate MHC-II Pan Receptors for Improved Predictability of Potential T Cell Epitopes"

_biology, 2023, doi:10.3390/biology12091265_

Round 1
Reviewer 1 Report
In this paper the authors describe the extensive comparative analysis of the performance of MHCII immunoprecipitating antibodies in the MAPPs assay, using 3 donors and 3 biological drugs. They provide evidence that the combination of a few selected antibodies may outperform a single pan-MHCII specific antibody in this assay and may therefore be advantagous in certain settings when sensitive antigenicity analysis is required.
Author Response
Thanl you for your constructive and positive review!
Reviewer 2 Report
The manuscript is well written and is of interest to the researchers working on human biologics for different diseases or drug repurposing. The conclusion is supported by the data provided. There is only one concern- the color of different combinations in the bar chart is not distinct mainly in DP, DP/DQ, DQ, DR/DP/DQ. Please select a different color scheme to make the differentiation more visible.
Author Response
Dear Reviewer,
Thank you for your comments. Fig. 1 and Fig. 5 were changed to improve the visibility and readibality of these legends:
- The font and color scheme of the legend was increased in size
- We changed the color scheme of some of the categories, and we think that even the minor components are now visible in the figure.
Thank you!
Reviewer 3 Report
The study design is appropriate to answer the aim. This study added to what is already in the topic. The methodology is rich and precisely presented. The data is presented in an appropriate way. The text in the results add to the data and it is not repetitive. Statistically significant results are clear. It is clear which results are with practical meaning. Results are discussed from different angles and placed into context without being overinterpreted.
The conclusions answer the aim of the study. The conclusions are supported by references and own results.
The limitations of the study are not fatal, but they are opportunities to inform future research.
The article is consistent within itself. There are no major flaws associated with this article.
However, there are some specific comments on weaknesses of the article and what could be improved:
Major points - none
Minor points
1. You can remove the instructions in the brackets next to abstract, key words, etc.
2. The references should be removed from the conclusions and moved to the discussion.
Author Response
Dear reviewer,
Thank you very much for your comments.
1. You can remove the instructions in the brackets next to abstract, key words, etc. - This was done, thank you for pointing this to us.
2. The references should be removed from the conclusions and moved to the discussion. - the two references that were in the conclusion were removed (as they were already mentioned earlier in the text).
Thank you!